# Mapping nonlinear receptive field structure in primate retina at single cone resolution

Jeremy Freeman[1,2]*[†], Greg D Field[3,4]*[†], Peter H Li[4][§], Martin Greschner[4,5], Deborah E Gunning[6], Keith Mathieson[6], Alexander Sher[7], Alan M Litke[7], Liam Paninski[8], Eero P Simoncelli[9][‡], EJ Chichilnisky[4,10][‡]

[1]Janelia Research Center, Howard Hughes Medical Institute, Ashburn, United States; [2]Center for Neural Science, New York University, New York, United States; [3]Department of Neurobiology, Duke University School of Medicine, Durham, United States; [4]Salk Institute for Biological Studies, La Jolla, United States; [5]Department of Neuroscience, University of Oldenburg, Oldenburg, Germany; [6]Institute of Photonics, University of Strathclyde, Glasgow, United Kingdom; [7]Institute for Particle Physics, University of California, Santa Cruz, Santa Cruz, United States; [8]Department of Statistics, Columbia University, Columbia, United States; [9]Center for Neural Science, Courant Institute of Mathematical Sciences, Howard Hughes Medical Institute, New York University, New York, United States; [10]Department of Neurosurgery, Stanford School of Medicine, Stanford, United States

*For correspondence:
freemanj11@janelia.hhmi.org
(JF); field@neuro.duke.edu (GDF)

[†]These authors contributed equally to this work

[‡]These authors also contributed equally to this work

Present address: [§]Google Research, San Francisco, United States

Competing interests: The authors declare that no competing interests exist.

**Abstract** The function of a neural circuit is shaped by the computations performed by its interneurons, which in many cases are not easily accessible to experimental investigation. Here, we elucidate the transformation of visual signals flowing from the input to the output of the primate retina, using a combination of large-scale multi-electrode recordings from an identified ganglion cell type, visual stimulation targeted at individual cone photoreceptors, and a hierarchical computational model. The results reveal nonlinear subunits in the circuitry of OFF midget ganglion cells, which subserve high-resolution vision. The model explains light responses to a variety of stimuli more accurately than a linear model, including stimuli targeted to cones within and across subunits. The recovered model components are consistent with known anatomical organization of midget bipolar interneurons. These results reveal the spatial structure of linear and nonlinear encoding, at the resolution of single cells and at the scale of complete circuits.

## Introduction

The responses of many sensory neurons reflect multiple stages of computational transformation, arising from the functional and anatomical organization of underlying neural circuitry. Although much of our knowledge of these circuits comes from direct measurements of responses during sensory stimulation, experimental and analytical techniques for inferring the properties of intermediate stages remain limited. In particular, in many neural circuits, the critical computations performed by interneurons are difficult to measure directly.

In the retina, cones, bipolar cells, and retinal ganglion cells (RGCs) form a two-tier cascade of linear and nonlinear processing that is thought to underlie many important visual functions (see *Schwartz and Rieke, 2011*; *Gollisch, 2012*). A notable example is the *nonlinear subunit* computation (*Hochstein and Shapley, 1976*) observed in the responses of several types of RGCs, the output neurons of the retina. In these cell types, the circuitry composing the RGC receptive field does not

**eLife digest** Light that enters the eye begins the process of vision by activating two types of photoreceptors: rods, which support vision under low light levels, and cones, which are responsible for fine detail and color vision. Activation of either type of photoreceptor triggers responses in bipolar cells, which activate the ganglion cells that transmit visual signals to the brain.

Bipolar cells therefore belong to a class of cells called interneurons, which relay information from certain cell types to others. Interneurons play an important role in information processing throughout the brain, but directly accessing them or characterizing their role in neural computation is often difficult.

To address this problem, Freeman, Field et al. have developed a combined computational and experimental approach to describe the flow of sensory signals through the circuits within the retina of primates. Large arrays of electrodes were used to record the responses of many retinal ganglion cells in response to the activation or de-activation of pairs of cones. These experiments revealed that the responses of ganglion cells are not simply the sum of the inputs that they receive from cones; specifically, the activation of one cone is not cancelled by the deactivation of another. Instead, the data suggest that bipolar cells add cone inputs together and then pass on the total activation (but not deactivation) to ganglion cells.

By analyzing the responses of ganglion cells to numerous random patterns of cone activation, Freeman, Field et al. were able to estimate the locations and arrangements of bipolar cells that connect to them. These predicted patterns of connectivity agreed with observations from anatomical studies. This work provides detailed insights into how the primate retina works. It also suggests that similar approaches may be used to characterize how signals flow across other brain networks in which large-scale recordings are now possible.

linearly integrate cone photoreceptor inputs over space, as assumed by simplified descriptive models (*Chichilnisky, 2001*; *Keat et al., 2001*; *Pillow et al., 2008*; *Field et al., 2010*). Instead, cone signals are first combined by bipolar interneurons, whose outputs are rectified at the synapse onto RGCs (*Demb et al., 1999*, *2001*; *Borghuis et al., 2013*). This nonlinear architecture endows the RGC receptive field with localized subunits that permit the representation of finer spatial detail than would be expected given the overall receptive field size (*Hochstein and Shapley, 1976*; *Lee et al., 1995*; *Demb et al., 2001*; *Baccus et al., 2008*; *Crook et al., 2008*; *Schwartz et al., 2012*). Subunits have also been implicated in the processing of features like object motion and looming (*Olveczky et al., 2007*; *Münch et al., 2009*), and some of their functional characteristics have been inferred for various cell types and species (*Enroth-Cugell and Robson, 1966*; *Hochstein and Shapley, 1976*; *Victor and Shapley, 1979a*, *1979b*; *Enroth-Cugell and Freeman, 1987*; *Bölinger and Gollisch, 2012*; *Schwartz et al., 2012*). However, the spatial organization of convergent cone input onto nonlinear subunits has never been visualized, nor mapped across a population of RGCs.

We combined large-scale parallel recordings of the responses of RGC populations to high resolution visual stimulation in order to characterize RGC subunits at the resolution of photoreceptors. Independent stimulation of individual cones and pairs of cones revealed significant nonlinear interactions within the receptive fields of OFF midget RGCs. A cascade model and an associated fitting method were developed to capture these nonlinear response properties. The model revealed the spatial structure of nonlinear interactions in the individual cone inputs to each OFF midget RGC, and enabled adaptive generation of validation stimuli targeted to specific cones during experiments. When fitted to entire populations of OFF midget RGCs, the model revealed a spatial organization of nonlinear subunits consistent with the known anatomical convergence of cones to midget bipolar cells.

## Results

RGCs in the primate retina (*Macaca mulatta* and *Macaca fascicularis*) were recorded and identified using a large-scale multi-electrode recording system and visual stimulation (*Dabrowski et al., 2004*; *Frechette et al., 2005*; *Field et al., 2007*). Inputs from individual cones to RGCs were mapped using high-resolution stimuli (*Field et al., 2010*). Experiments and analyses focused on the responses of OFF midget cells over a range of eccentricities. These cells exhibited the largest signal-to-noise ratios and response stability over recordings.

## Primate OFF midget RGCs exhibit nonlinear spatial integration

Targeted stimulation of cone pairs within the receptive field of RGCs revealed nonlinear interactions. Specifically, increments and decrements of light were presented in regions that illuminated individual cones or pairs of cones (*Figure 1*, see 'Materials and methods'). If a RGC linearly combines cone inputs, its response should be eliminated, or strongly reduced, when inputs of opposite polarity are presented to two separate cones that are similarly weighted within the receptive field. To test this prediction, we selectively stimulated pairs of cones that were close in space, but sufficiently distinct to ensure independent stimulation (see 'Materials and methods').

Many cells exhibited non-linear responses to opposite polarity stimulation of cone pairs (*Figure 1A,B*). Specifically, many OFF midget cells responded robustly to decrements of light presented to either of a pair of cones individually, but also responded robustly when a decrement of light was presented to one cone and an increment of the same contrast was presented simultaneously to the other cone (for 73/78 cone pairs tested from 13 RGCs, response to paired stimulations was at least 50% of the response to single cone decrements alone). Only a small number (5/78) of cone pairs yielded response cancellation (see example in *Figure 1C*). These failures of cancellation indicate significant spatial nonlinearities within OFF midget cell receptive fields.

These examples are consistent with the following interpretation suggested by preceding work in other species and cell types: OFF midget bipolar cells (which carry cone signals to OFF midget RGCs) (*Kolb and Marshak, 2003*) linearly combine convergent cone input, but introduce a rectifying nonlinearity at the synapse with RGCs (*Demb et al., 2001*; *Bölinger and Gollisch, 2012*; *Schwartz et al., 2012*). In this interpretation, the cancellation of opposing signals from a pair of cones indicates that the cones converge on a single midget bipolar cell, while failure of cancellation indicates that the cones provide input to distinct midget bipolar cells. The analyses and experiments described below use a model-based approach that provides further evidence for this interpretation.

## Inferring cone signal interactions with a hierarchical subunit model

The paired stimulation approach described above is not suitable for characterizing large populations of cones and RGCs, because it relies on testing a small number of cone pairs, in a small number of RGCs per retina. Testing all cone pairs in all RGCs is infeasible. It is possible, however, to identify the two key properties of interest—the presence of an intermediate nonlinearity, and the specific pairs (or groups) of cone signals that combine linearly in a common subunit—by fitting a single hierarchical model to the responses of RGCs under random stimulation with spatiotemporal noise. Across many random stimulus presentations, different spatial patterns will engage different degrees of linear and nonlinear spatial integration within the receptive field. Thus, finding the model that best predicts the RGC response can, in principle, reveal the structure of the nonlinearities.

We developed a hierarchical model with two stages of linear integration and an intervening static nonlinearity (*Figure 2*) mirroring the known elements of the underlying circuitry (*Figure 2*). First, subsets of one or more cones are assigned to 'subunits', which are assumed to be non-overlapping to ensure unique solutions during fitting, that is, a cone cannot provide input to multiple subunits (see 'Materials and methods'); this assumption is biologically sensible given evidence for minimal overlap in bipolar dendritic trees (*Wässle et al., 1994*) and little divergence from cones to OFF midget bipolar cells (*Kolb and Marshak, 2003*). Signals from cones within a subunit are weighted and linearly combined. Second, a common nonlinear transformation is applied to the outputs of these subunits (*Ahrens et al., 2008*). Finally, the model computes a weighted sum of subunit responses, meant to reflect the convergence of bipolar cells onto the ganglion cell, and this summed response is subjected to an output nonlinearity that produces a non-negative spike rate. This model builds on previous functional models of nonlinear spatial integration in RGCs (*Victor and Shapley, 1979a*, *1979b*; *Enroth-Cugell and Freeman, 1987*) by describing the subunit computation directly with components that resemble elements of the circuit, and by making predictions about their organization at the resolution of individual cones. It also bears resemblance to two-stage subunit models of V1 complex cells (*Adelson and Bergen, 1985*; *Rust et al., 2005*; *Chen et al., 2007*), and recently developed methods of fitting such models to data (*Ahrens et al., 2008*; *Vintch et al., 2012*; *Lochmann et al., 2013*; *McFarland et al., 2013*).

The model was fitted to hundreds of simultaneously recorded OFF midget RGCs by measuring responses to high-resolution spatio-temporal noise (*Field et al., 2010*) (15 retinas total, see 'Materials

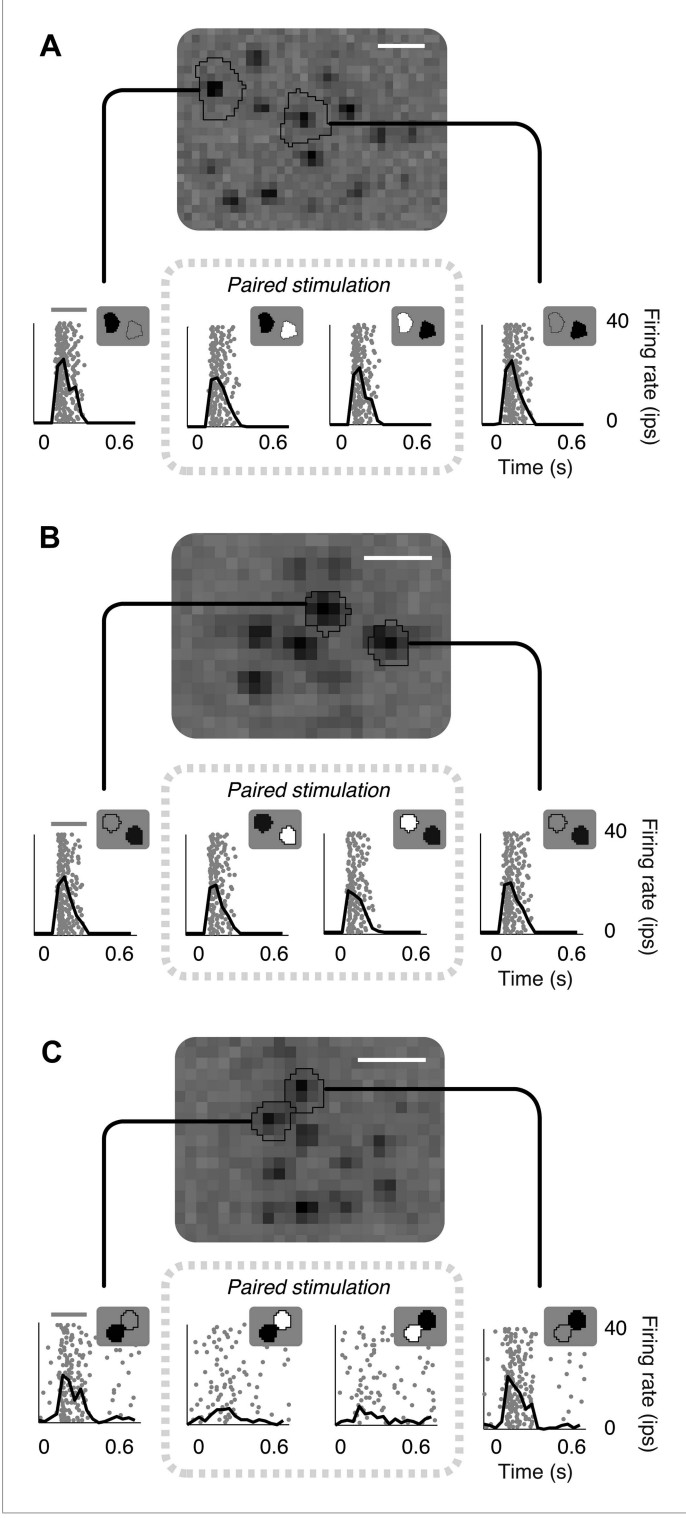

**Figure 1**. Failure of linear integration in OFF midget retinal ganglion cells (RGCs). (**A**) Above, spatial spike-triggered average derived from white-noise analysis with high-resolution pixel stimulation. Black lines indicate regions of pixels independently stimulating individual cones (identified online). White line, scale bar (8.4 microns). Below, rasters of responses to repeated brief presentations of uniform luminance within each cone region. Each row is a trial, each point is a spike. Black line, average firing rate across trials. Gray line, stimulus presentation (250 ms). Stimulation in cone regions was either paired increments and decrements of light, or decrements alone, as shown in insets. (**B**) Another RGC showing failure of cancellation. (**C**) A RGC exhibiting cancellation.

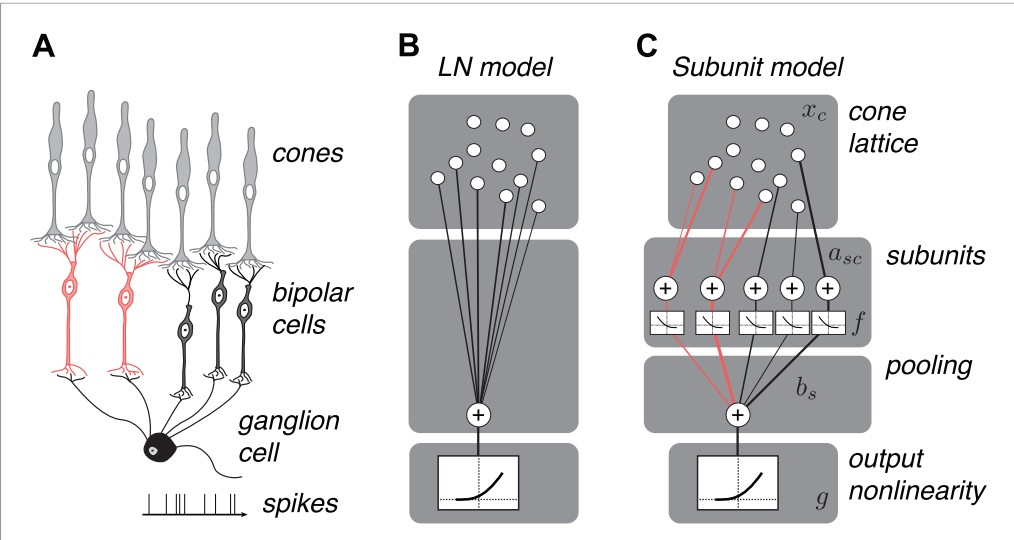

**Figure 2**. OFF midget RGC anatomy and modeling framework. (**A**) Bipolar cells receive convergent input from cones, and ganglion cells receive convergent input from bipolar cells. (**B**) A linear–nonlinear (LN) model describes the RGC response with one stage of linear integration of cone inputs, followed by a nonlinearity. White disks indicate cones, black line indicates integration weights. (**C**) The subunit model describes the RGC response with two stages of linear integration and nonlinearity; subunits perform initial integration following by a nonlinearity (common to all subunits). Red, subunits receiving input from two cones; black, subunits receiving input from one cone. Cones are indexed by $c$, subunits by $s$. $x_c$ is the input to each cone. $a_{sc}$ is the weight from cone $c$ to subunit $s$. $b_s$ is the weight on each subunit. $f$ and $g$ are the subunit and spiking nonlinearities, respectively. See 'Materials and methods' for details on parameterization and fitting.

and methods'). First, cone locations were identified using a previously described fitting procedure (*Field et al., 2010*) (see 'Materials and methods'), and the noise stimulus was then spatiotemporally filtered for each cell to describe the stimulus in terms of cone inputs rather than pixel intensities. Second, model parameters (the two nonlinearities, and the summation weights over the cones and over the subunits) were adjusted to maximize the likelihood of the spiking responses of each cell, assuming spikes arose from an inhomogeneous Poisson process (*Simoncelli et al., 2004*; *Pillow et al., 2008*). Specifically, coordinate ascent was used to infer the weights and nonlinearities at both stages of the model, and the assignment of cones to subunits was determined by an iterative greedy merging method, initializing to a single-cone subunit model (one cone per subunit), and then iteratively merging the pair of subunits that yielded the largest improvement in response likelihood (see 'Materials and methods').

## Fitted models recover nonlinearities and subunit organization

The fitted functional model for each cell is depicted graphically (*Figure 3A*). For most cells, the estimated subunit nonlinearity approximated half-wave rectification, consistent with the failures of cancellation observed in paired stimulation. Subunits were generally small. Many OFF midget RGCs had only single-cone subunits (48 ± 10%, percentage ±s.e.m. across retinas). The remaining cells generally had subunits with two (27 ± 6% OFF midgets) or three (13 ± 4% OFF midgets) cones. The performance of the model was assessed by computing the percent of firing rate variance explained ($R^2$) on a portion of the data not used in the fitting process (see 'Materials and methods'). Although the model was fitted by maximizing likelihood under a Poisson spiking model, $R^2$ provides a more easily interpretable measure of performance. Model improvement was assessed relative to a standard linear–nonlinear (LN) cascade model (*Chichilnisky, 2001*), containing only one stage of linear integration followed by a nonlinearity, which was fitted and evaluated using identical methods. Results from each retina were summarized with the average nonlinearities (see *Figure 3B* for example subunit nonlinearities), as well as the improvement in predictive accuracy of the subunit

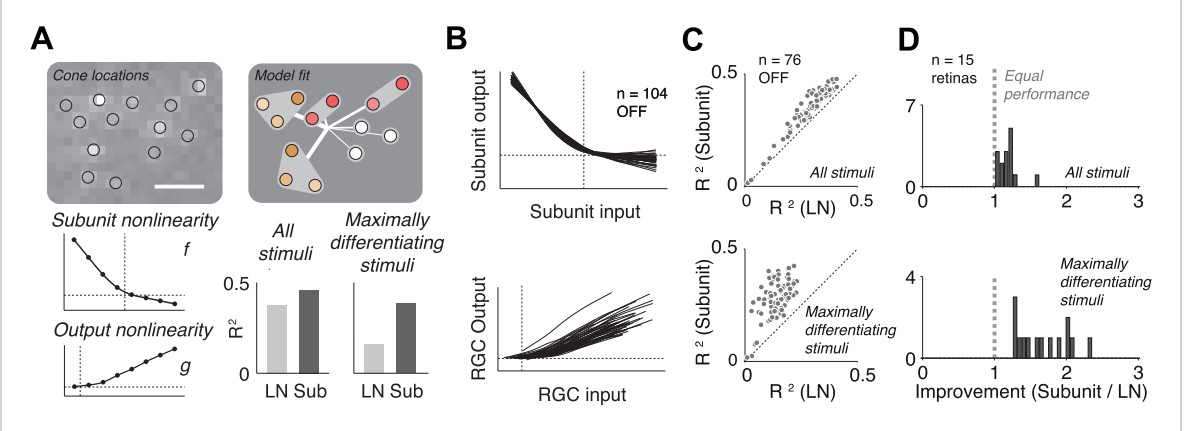

**Figure 3**. Model fits. (**A**) Recovered model fit for an example OFF midget RGC. Top left, spatial spike-triggered average. Identified cone locations indicated by black circles. White line, scale bar (8.4 microns). Top right, diagram of model fit. Light gray regions indicate subunits, including those containing single cones. Within a subunit, color saturation indicates the weight on each cone. Red indicates that the subunit has two cones, orange, three. Line intensities indicate weights used in summing over subunits. Bottom left, recovered subunit and output nonlinearity. Bottom right, explanatory power ($R^2$, see 'Materials and methods') of LN and subunit ('Sub') models. $R^2$ was computed either for all presented stimuli, or for maximally differentiated stimuli, defined as those stimuli for which the predictions of the two models differed the most (see 'Materials and methods'); in both cases models were evaluated on data not used for model fitting. (**B**) Summary of subunit and output nonlinearities for a population of 104 OFF midgets from one retina. Each line corresponds to a single RGC. Nonlinearities only shown for RGCs with $R^2$ exceeding 0.2. (**C**) Summary of model performance for 76 OFF midget RGCs from an example retina. Above, $R^2$ for subunit and LN models on all stimuli; below, $R^2$ for subunit and LN models on maximally differentiating stimuli (see panel **A**, and 'Materials and methods'). Each corresponds to a single RGC. (**D**) Histogram of model improvement (subunit/LN) across multiple retinas, quantified for each retina as the slope of the best-fitting regression line to the points shown in panel **C** (forced through the origin). The data point for OFF midgets from one outlying retina (improvement = 5) is not shown.

The following figure supplement is available for figure 3:

**Figure supplement 1**. In three separate simulations, spiking responses were simulated using a known set of parameters (derived from the model fit to a real OFF midget RGC), and simulation duration was either 4, 8, or 16 min (typical experiments were 30 min).

model over the LN model (percent increase in cross-validated $R^2$, quantified as the slope of the best-fitting regression line relating the performance of the two models, *Figure 3C*). Note that the subunit model can mimic an LN model when all subunits contain single cones and the subunit nonlinearity is linear, and in this case, it cannot underperform the LN model unless the data are insufficient to constrain it (i.e., 'overfitting'). We found an improvement of 18 ± 3% (mean ± s.e.m.) across 15 retinas (*Figure 3D*).

The improvement in predictive power of the subunit model over the LN model was modest, in part because the predictions were performed over a large ensemble of random noise stimuli that only infrequently and weakly activate cones and subunits in a manner that differentiates the model predictions. To obtain a more incisive comparison, for each fitted cell, the model (fitted to training data) was used to identify stimuli (from data not used in fitting) for which the predictions of the two models differed the most. Because independent data were used for training and testing, and because the selection of 'potent' stimuli was based only on the magnitude of the difference in predicted response, the improvement is not statistically biased in favor of one model or the other. Model accuracies on these 'maximally differentiating' test stimuli revealed a larger improvement of 92 ± 26% (*Figure 3C*). Variation in improvement across retinas (*Figure 3D*) may have arisen from differences in physiological state across tissue preparations (see 'Discussion').

In order to assess model accuracy, responses were averaged over repeated presentations of a short spatio-temporal noise sequence (*Figure 4A*). These were used to assess the accuracy of predictions of both the subunit and LN models, expressed as a fraction of the stimulus-induced variation of the response over time ($R^2$ adjusted, see 'Materials and methods'). Accuracy for the subunit model was 51 ± 1% $R^2$ adj. (mean ± s.e.m. across 68 OFF midget RGCs from one retina), and improvement over the LN model was 40% (*Figure 4B*).

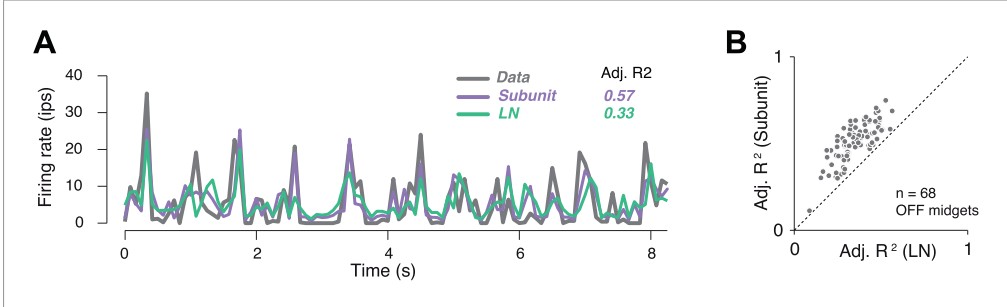

**Figure 4**. Subunit model predicted responses to repeated white noise. (**A**) Average firing rate of a single OFF midget RGC to an 8 s white noise stimulus repeated 100 times. Purple, prediction of subunit model; green, prediction of LN model. Both models were fit to independent data using non-repeated white noise stimuli. Bin width, 83 ms. (**B**) Adjusted $R^2$ (see 'Materials and methods') for the two models; each point shows performance for one OFF midget RGC.

## Subunit model more accurately predicts responses to grating stimuli

A traditional test of nonlinear integration involves measuring responses to contrast reversing sinusoidal gratings (*Enroth-Cugell and Robson, 1966*; *Victor and Shapley, 1979a*, *1979b*; *Enroth-Cugell and Freeman, 1987*). Responses to gratings were measured in 32 OFF midget RGCs from one retina. RGCs exhibited clear signs of nonlinear integration (see *Figure 5A* for an example cell). At low spatial frequencies (period matched approximately to the size of the receptive field), responses were modulated at the temporal frequency of the stimulus. But at higher spatial frequencies, their responses were modulated at twice the temporal frequency of the stimulus, for at least some spatial phases. The rectifying nonlinearities within the fitted subunit model were sufficient to predict this frequency doubling, whereas the LN model exhibited dominant response modulation at the frequency of the stimulus (*Figure 5A*). Across the population, accuracy of the subunit model was 83 ± 1% $R^2$ adj. (mean ± s.e.m. across 32 RGCs), and improvement in predictive performance of the subunit model over the LN model was 28 ± 2% (mean ± bootstrapped confidence interval) (*Figure 5B*). Improvement was more pronounced at higher spatial frequencies (*Figure 5C*), ranging from 10 ± 1% (at a frequency where one cycle spanned the entire RF) to 50 ± 6% (at a frequency where each cycle spanned 1–2 cones), and was significant at all frequencies (paired t-test, p < 0.0001). A qualitatively similar pattern of results was observed in a second retina (data not shown). Especially for the higher spatial frequencies, the difference between models for gratings was qualitatively more pronounced than for noise stimuli (*Figure 3C*), suggesting that stimuli with extended spatial structure help highlight differences between model predictions.

## Subunit structure validated anatomically and functionally

Although the subunit model provided large improvements over an LN model, the improvement of the full fitted subunit model over a simpler model with all subunits as single cones was small: roughly 0–10% across all RGCs (see 'Materials and methods'). The small difference was likely due to the small size and number of multi-cone subunits per RGC—typically no more than a few subunits with two or three cones—and the simplistic spatial structure of the stimuli used. This raises the question of whether the multi-cone subunits inferred from model fits truly reflect the function and anatomy of the underlying neural circuitry.

Anatomical measurements of synapses between cones and OFF midget bipolar cells (*Wässle et al., 1994*) provide direct estimates for the expected number of cones that synapse on a bipolar cell as a function of retinal eccentricity. The functional connectivity of the fitted model provides, for each recorded retina, a prediction of the number of cones per bipolar cell (*Figure 6*). This allowed a comparison between the anatomical connectivity and the functional connectivity predicted by the full subunit model. Note that comparing the number of subunits rather than cone convergence onto subunits is approximately equivalent, because midget bipolar cells tile the retina with a coverage

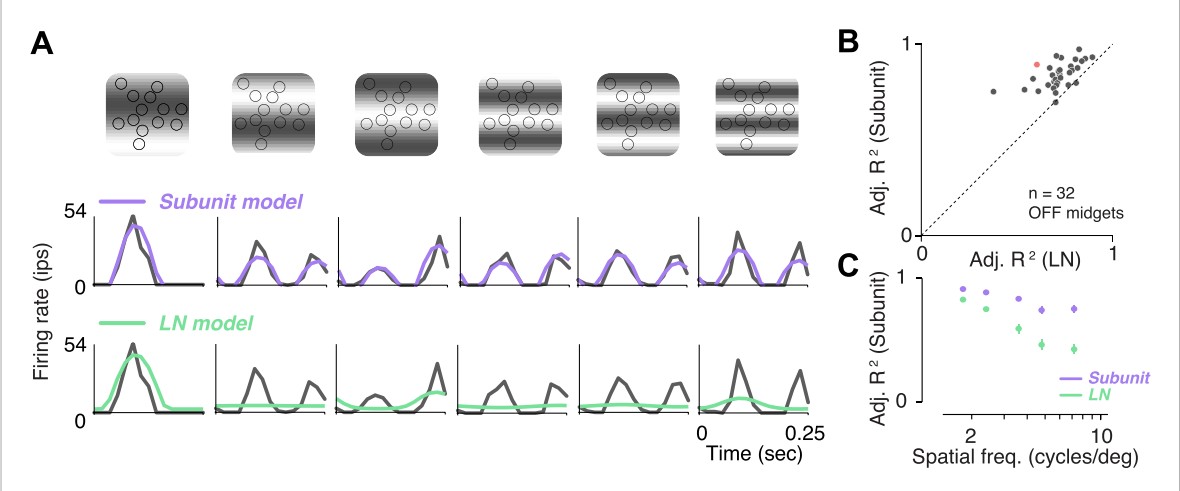

**Figure 5**. Responses to gratings. (**A**) Sinusoidal gratings temporally modulated in contrast were presented to OFF midget RGCs. Top: grating stimuli superimposed on the cone locations of a single RGC, for a set of example combinations of spatial frequency and phase (with spatial frequency increasing from left to right). Bottom: cycle-averaged firing rate of the RGC (black); predictions of the LN model (green); predictions of the subunit model (purple). Both models were fit to independent data from white-noise stimulation. (**B**) Performance (adjusted $R^2$, see 'Materials and methods') for subunit and LN models, across 32 OFF midget RGCs from a single retina. Data point in red corresponds to example shown in panel **A**. (**C**) Adjusted $R^2$, computed separately for the two models for different spatial frequencies, averaged across RGCs. Error bars indicate s.e.m. across RGCs.

factor of ~1 and without overlap (*Wässle et al., 1994*). For RGCs at small eccentricities, the fraction of subunits recovered by the model including 1–2 cones was consistent with the fraction expected based on anatomical observations made separately (*Figure 6D*). For larger eccentricities, the model recovered larger subunits, also generally in agreement with anatomical observations. Two inconsistencies include some unexpectedly large subunits at high eccentricities (*Figure 6C*, [*Wässle et al., 1994*]) and one surprising distribution of subunit sizes at high eccentricity (*Figure 6D*). Despite this, the data generally support the interpretation that the fitted model revealed the structure of interneuron circuitry responsible for RGC responses.

To validate the model fits functionally, we developed a closed-loop experiment to test the specific subunit structure of individual cells recovered by the model. The subunit model was fitted to many OFF midget RGCs simultaneously based on responses to noise stimuli, and these fits were then used to guide the selection of RGCs and cone pairs to be tested for linear cancellation. This is identical to the experiment presented earlier (*Figure 1*), with the important difference that cone pairs were chosen based on predictions from the fitted model, during the experiment, instead of at random. Responses to combined increment and decrement stimulation were qualitatively consistent with the predictions of the model: For cones within subunits, RGC responses to inputs of opposite polarity were at least partially cancelled, whereas responses to cones in separate subunits exhibited little cancellation (*Figure 7*). Quantitatively, across 252 cone pairs tested (from 21 RGCs), the responses were significantly more accurately predicted by the full subunit model ($r^2 = 0.24$) than by either a model with single-cone subunits ($r^2 = 0.11$) or an LN model ($r^2 = 0.10$) (both p < 0.005, bootstrap test, see 'Materials and methods').

## Discussion

We used a combination of measurement and modeling to reveal nonlinear interneuron computations underlying the responses of RGCs in primate retina. By presenting stimuli at the resolution of individual cones, and fitting a cascade model to the spiking responses of OFF midget RGCs, we identified individual nonlinear subunits that are consistent with the known anatomical convergence from cones to midget bipolar cells. We fitted and validated the model using a comprehensive stimulus ensemble, including noise, gratings, and targeted stimuli that highlighted the nonlinear responses. Together, these results provide a picture of the spatial circuit structure of three major stages of visual processing in the primate retina.

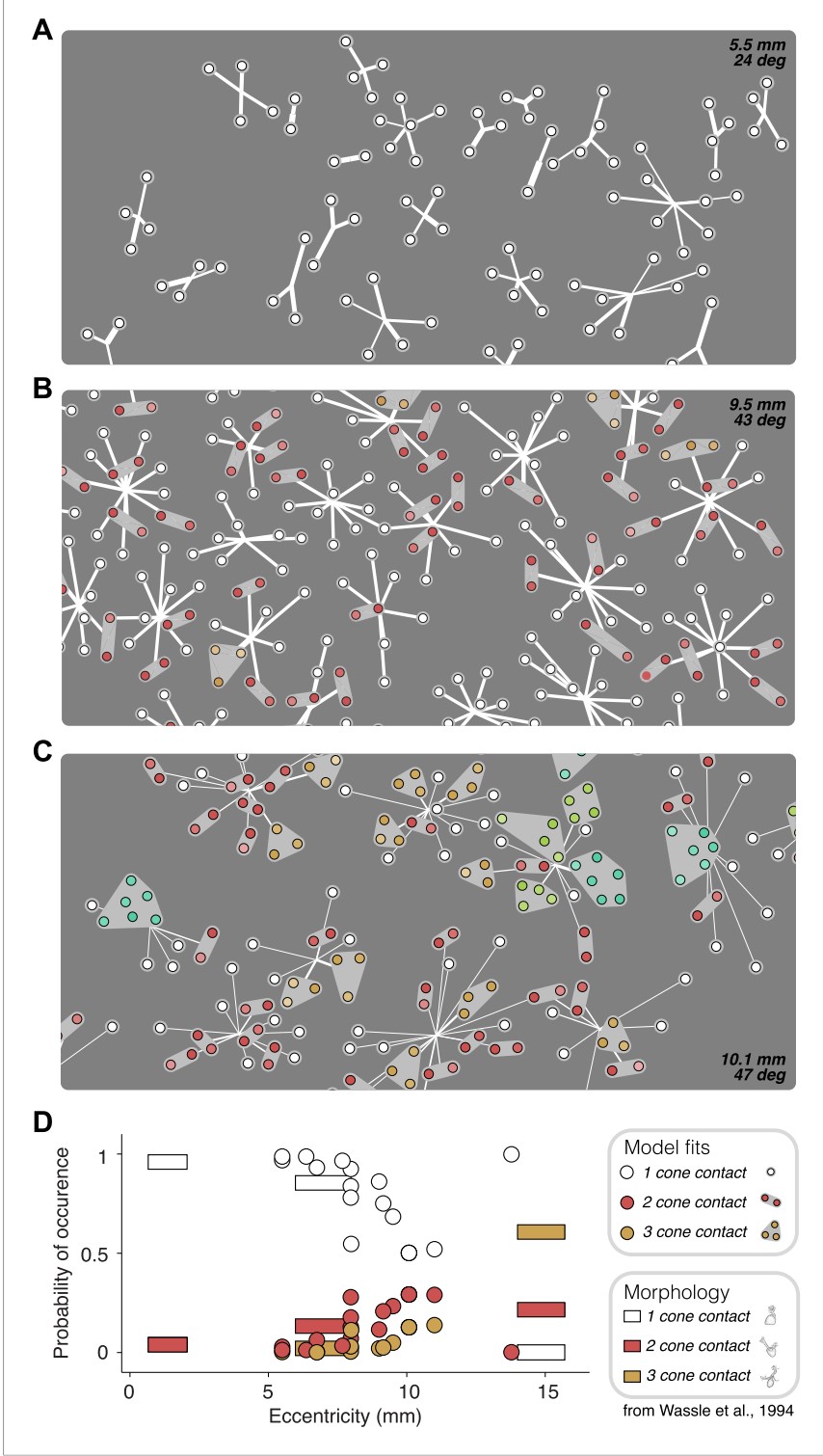

**Figure 6**. Recovered subunit structure varies with eccentricity. (**A–C**) Model fits recovered for local populations of OFF midget RGCs from retinas at three different retinal eccentricities. Model fits for each RGC depicted as in *Figure 3*. Colors indicate the number of cones in each subunit; white = 1, red = 2, orange = 3, lime = 4, turquoise = 6. (**D**) Probability of occurrence of subunits with different numbers of cone contacts as a function of eccentricity, for morphological and functional data. Horizontal bars show estimates derived from morphological data from *Wässle et al., 1994*. Circles show estimates from the functional measurements and model fitting for individual retinas.

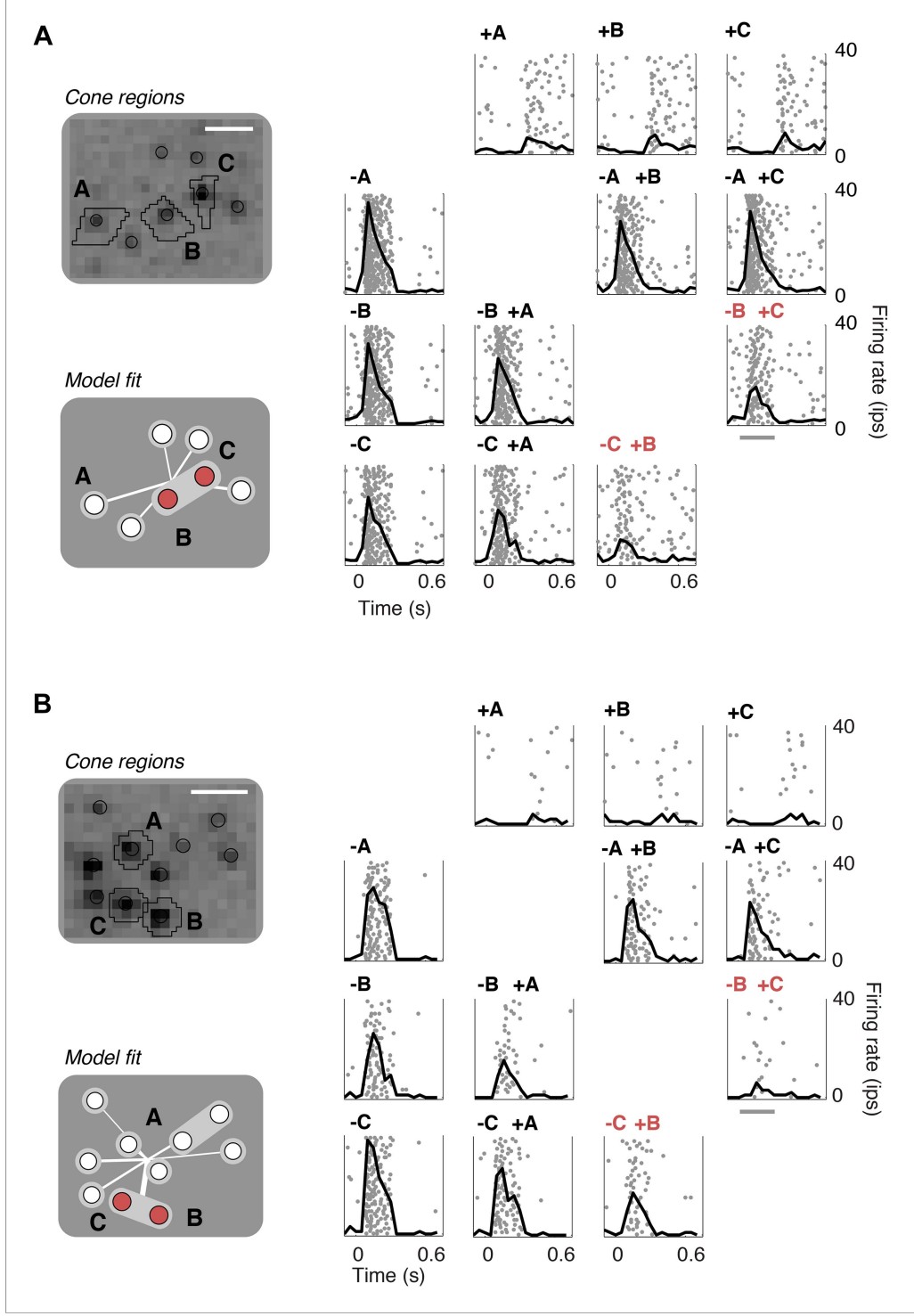

**Figure 7**. Closed-loop validation of subunit organization. (**A**) Upper left, spatial spike-triggered average and cone stimulation regions, as in *Figure 1*. Lower left, diagram of fitted subunit model, as in *Figure 3A*. The three cones (A, B, C) were stimulated with increments (+) or decrements (−) of light. Panels on the right show the responses (as in *Figure 1*) of a single OFF midget RGC to different combinations of light increments and decrements. Top row, and left column, show responses to stimulation of single cones. Remaining panels show responses to paired stimulation. Red indicates the pair of cones found by the model to belong to a single subunit, and thus selected for paired stimulation. Gray line, stimulus presentation (250 ms). (**B**) Another example, plotted as in panel **A**.

## Nonlinear spatial summation in OFF midget RGCs

Previous work has aimed to identify whether different RGC classes exhibit linear or nonlinear spatial summation. In the cat retina, 'X' cells were defined as those that sum visual input linearly within the spatial receptive field, whereas 'Y' cells were defined as those that exhibited nonlinear behaviors (notably, frequency-doubling) (*Enroth-Cugell and Robson, 1966*; *Hochstein and Shapley, 1976*). These functionally defined classes have been associated with the morphologically distinct beta and alpha RGCs (*Boycott and Wässle, 1974*; *Peichl and Wässle, 1981*; *Wässle et al., 1981a*, *1981b*). Subsequent work aimed to establish a similar dichotomy in the primate retina (*Field and Chichilnisky, 2007*). The magnocellular-projecting parasol RGCs have been shown to exhibit nonlinearities similar to Y cells. In contrast, the parvocellular-projecting midget RGCs have usually been likened to linear X cells (*Shapley et al., 1981*; *Kaplan and Shapley, 1982*; *Derrington and Lennie, 1984*; *Levitt et al., 2001*). The nonlinearities revealed here (*Figures 1, 4, 6*) are inconsistent with this interpretation, as are previous in vivo studies of parvocellular-projecting cells (*Derrington and Lennie, 1984*) and ex vivo studies of midget RGCs (*Petrusca et al., 2007*; *Cafaro and Rieke, 2013*), which demonstrated modest but clear frequency-doubling in response to contrast reversing visual stimuli. Two factors may explain these discrepancies. First, many previous in vivo studies probed midget cell responses in the central retina, where the receptive field center consists of a single cone, and thus, a single subunit. In such cases, the entire receptive field is formed by the cone collecting aperture, and would be expected to exhibit the linear summation of incident light intensity performed by the cone. Second, in vivo studies were typically performed using anesthetics that could raise the basal release of neurotransmitter from bipolar cells, and low basal release rates are thought to underlie at least part of the rectifying subunit nonlinearity (*Robbins and Ikeda, 1989*; *Demb et al., 2001*). Indeed, variability in basal release rates could account for some of the variability across preparations observed here (*Figures 3D, 6*).

Although OFF midget RGCs exhibit nonlinear spatial summation not found in X cells, they are also different from Y cells. The most commonly stated defining characteristic for Y cells is a frequency doubled response to a contrast reversing grating, with a magnitude that is independent of the spatial phase of the stimulus (*Enroth-Cugell and Robson, 1966*; *Hochstein and Shapley, 1976*; *Crook et al., 2008*). Although midget cells exhibited frequency doubling in response to contrast reversing gratings (*Figure 5*), the magnitude was phase dependent (not shown). Phase dependence may arise both from the low occurrence of multi-cone subunits, and the fact that midget cells receive input from only one bipolar cell type, which forms a single mosaic with minimal dendritic overlap (*Wässle et al., 1994*). In contrast, the ON and OFF parasol RGC populations, which exhibit properties consistent with the Y cell definition (*Petrusca et al., 2007*; *Crook et al., 2008*), receive input primarily from two bipolar cell types (*Bordt et al., 2006*; *Calkins and Sterling, 2007*), each of which receive convergent input from many cones, and overlap each other in space, and thus would be expected to exhibit greater phase invariance. Lack of phase independence in midget cells may also arise from differences between feed-forward and crossover inhibition that shapes midget vs parasol RGC responses respectively (*Cafaro and Rieke, 2013*).

A final potential caveat in comparing our findings to previous work is the possibility that the ex vivo preparation used here affected the properties of midget cells. The firing rates obtained in the present work are somewhat lower than those found in anesthetized in vivo recordings (*Troy and Lee, 1994*). Nonlinear spatial summation (as in *Figure 1A,B*) was observed in midget cells exhibiting a range of spontaneous firing rates from 0 to 10 spikes/s (not shown). It remains uncertain whether and to what degree these ex vivo recordings, or in vivo recordings under anesthesia, reflect a more natural state with respect to the signals studied. The confirmation of the biological relevance of nonlinear signaling by OFF midget cells awaits a decisive test in awake behaving animals.

## Modeling subunit structure in RGCs

The model developed here was inspired by previous subunit models (*Victor and Shapley, 1979b*), general successes in maximum likelihood fitting of neural models to spiking responses (*Paninski, 2004*; *Truccolo et al., 2005*; *Ahrens et al., 2008*; *Pillow et al., 2008*), and recent efforts to fit subunit-like models to V1 complex cells (*Rust et al., 2005*; *Vintch et al., 2012*; *Lochmann et al., 2013*; *McFarland et al., 2013*). But it differs in several essential aspects. First, the model expresses subunit computations directly in terms of elementary circuit components: photoreceptors, inferred bipolar

cells, and ganglion cells. Combined with precise experimental control, this approach bridges the gap between abstract functional models and detailed circuit computations, and may thus complement the growing number of tools available for manipulating neurons and circuits. Second, unlike some previous efforts to characterize subunits (*Victor and Shapley, 1979b*), all components of the model used here were fitted based on a single ensemble of stimuli and spiking responses, rather than inferred separately based on responses to specialized stimuli. Simultaneous fitting is a generally preferable strategy for optimization, because it allows the parameters to work together to explain observed responses. Finally, unlike some approaches (e.g., spike-triggered covariance [*Strong et al., 1998*; *Brenner et al., 2000*; *Rust et al., 2005*; *Touryan et al., 2005*; *Pillow and Simoncelli, 2006*; *Schwartz et al., 2006*; *Chen et al., 2007*] or maximally-informative dimensions [*Sharpee et al., 2004*]), which have recovered stimulus features that are unique only up to an arbitrary linear transformation, this method recovers a unique solution for the spatial structure of subunits.

Two recent efforts characterized RGC nonlinearities using novel experimental and anatomical methods that complement the approaches described here. One study (*Bölinger and Gollisch, 2012*) used a closed-loop experiment to find combinations of stimuli eliciting constant RGC responses, and used the shapes of the resulting iso-response contours in stimulus space to dissociate linear and nonlinear integration. While elegant, iso-response contour mapping has only proven practically useful with stimuli defined in two or three dimensions (*Bölinger and Gollisch, 2012*; *Horwitz and Hass, 2012*), whereas the current model operated on the full set of cone inputs to each RGC (typically 5–15 dimensions). Another study (*Schwartz et al., 2012*) developed a model for responses of ON alpha-like mouse RGCs by combining detailed morphological measurements with direct physiological measurements from bipolars and RGCs. The functional approach taken here, while lacking the firm anatomical foundation, can extend more readily to describing computations in complete neuronal populations (*Figure 6*) and in multiple cell types.

The model developed here does not account for electrical coupling between photoreceptors (*Tsukamoto et al., 1992*; *Hornstein et al., 2004*; *O'Brien et al., 2012*) which could produce cancellation of increment and decrement stimuli delivered to neighboring cones. A study of cone electrical coupling in primate retina suggested that signal mixing is modest (*Hornstein et al., 2004*), thus it is probably unable to account for the cancellation observed. However, that study was performed in dark-adapted retina, whereas the results reported here were obtained in light-adapted retina; coupling may depend on light adaptation and the circadian cycle (*Bloomfield and Völgyi, 2009*) (but see *Hornstein et al., 2004*). Thus, it is difficult to entirely rule out a contribution of cone coupling. However, since many cone pairs tested exhibited little or no cancellation (e.g., *Figure 1*), any effects of adjacent cone coupling are likely to be weak or sporadic. In contrast to this prediction, connexin-36 immunolabeling revealed ubiquitous expression at junctions between nearly all neighboring cones, in both central and peripheral primate retina (*O'Brien et al., 2012*). Also, the observed eccentricity dependence of the spatial extent of cancellation (*Figure 6*) implies that for the present results to be explained by cone coupling, the coupling would have to increase substantially with eccentricity.

## Applications and extensions of the subunit model

The model described here could potentially capture nonlinear computations in other RGC types, such as ON midget or parasol RGCs. But in some cases the model would require modification. For example, unlike midget RGCs, parasol RGCs receive input from more than one type of bipolar cell, and the receptive fields of these bipolar cells likely exhibit substantial spatial overlap, whereas the current model assumes nonoverlapping subunits (*Figure 2*). Indeed, preliminary model fits of parasol responses produce subunits smaller than what is expected from anatomical data, as would be expected in the case of two overlapping bipolar populations (*Boycott and Wässle, 1991*) (not shown). Subunit overlap may underlie the reported Y-cell invariance to spatial phase (see above), and is thus an important goal for future modeling.

Extensions to the model would also be required to capture RGC responses under more natural stimulus conditions. For example, the current model does not include receptive field surrounds. Insofar as surrounds reflect the contributions of horizontal and amacrine cells (*Mangel, 1991*; *Cook and McReynolds, 1998*; *McMahon et al., 2004*; *Ichinose and Lukasiewicz, 2005*; *Davenport et al., 2008*), additions to the model could capture the contribution of these interneuron types and elucidate their computational role. Incorporating inhibitory interneurons could also improve the accuracy of

subunit estimation, if other components in the current model are indirectly absorbing their effects. In some stimulus regimes, it may also be important to incorporate nonlinearities in cone signals. Specifically, the logic of the linearity test (*Figure 1*), and the structure of the model (*Figure 2*), assume linearity of the cone signal, which is a reasonable approximation for weak stimuli (*Schnapf et al., 1990*), but fails for stronger stimuli (J Angueyra and F Rieke, personal communication). This may account for the moderately lower predictive accuracy of the model when fit to noise and tested on stronger increment-decrement stimuli (*Figure 7*). The model would also be enhanced by allowing for temporal nonlinearities (rather than static ones) (*Borghuis et al., 2013*), post-spike temporal filtering (*Pillow et al., 2008*), more complex spike generation, and perhaps nonlinearities that vary across subunits. More generally, the model should be extended to include mechanisms for luminance and contrast gain control (*Mante et al., 2008*), both of which are essential for natural vision. All of these extensions would require new targeted stimulus ensembles and model fitting procedures, and together provide an exciting direction for future research.

The approach developed here also offers potential opportunities for uncovering the structure and function of other neural circuits. Subunit models have been used to explain pooling behaviors in primary and secondary visual cortex (*Hubel and Wiesel, 1959*; *Rust et al., 2005*; *Vintch et al., 2012*; *Freeman et al., 2013*; *Lochmann et al., 2013*), and may represent a 'canonical' computation of sensory processing (*Fukushima, 1980*; *Douglas et al., 1989*; *Heeger et al., 1996*; *Riesenhuber and Poggio, 1999*). New stimulation and measurement techniques that facilitate experimental manipulation of these circuits are proliferating. For example, optogenetics enables simultaneous stimulation of many neurons of a given class, while measuring electrical responses of other neurons (*Papagiakoumou, 2013*). But in most scenarios, the circuits of interest will contain neurons that are neither directly stimulated nor measured, analogous to the bipolar interneurons in the present work. Thus, further development of the computational approaches described here may prove an essential tool for inferring the function and connectivity of interneurons in neural circuits.

## Materials and methods

### Animals and dissections

Preparation and recording methods have been described previously (*Chichilnisky and Kalmar, 2002*). Eyes were enucleated from 15 terminally anesthetized macaque monkeys (*M. mulatta* and *M. fascicularis*) used by other experimenters in accordance with institutional guidelines for the care and use of animals. Five had been used for behavioral and neurophysiological experiments, eight had been experimentally exposed to ethanol (*Wright et al., 2013*), and two had been terminally anesthetized for several days. No systematic differences in retinal physiology were observed between categories of animals. Immediately after enucleation, the anterior portion of the eye and the vitreous were removed in ambient indoor illumination. Segments of peripheral retina (eccentricity 5.5–13.8 mm temporal equivalent [*Chichilnisky and Kalmar, 2002*]) that were well attached to the pigment epithelium were dissected and placed flat, RGC side down, on a planar array of extracellular microelectrodes. The array consisted of 512 electrodes in an isosceles triangular lattice with 30 μm spacing, covering a hexagonal region 450 μm on a side (*Field et al., 2010*). Attachment to the pigment epithelium was preserved during dissection. In some preparations the choroid was largely removed, up to Bruch's membrane, to ensure even retinal thickness and maximize oxygenation. While recording, the retina was perfused with Ames' solution (31–36°C) bubbled with 95% $O_2$ and 5% $CO_2$, pH 7.4.

### Spike sorting

Recordings were analyzed online to isolate the spikes of different cells, as described previously (*Dabrowski et al., 2004*; *Field et al., 2007*). Briefly, candidate spike events were detected using a threshold on each electrode, and voltage waveforms on the electrode and nearby electrodes around the time of the spike were extracted. Clusters of similar spike waveforms were identified as candidate neurons if they exhibited a 1 ms refractory period and accounted for more than 100 spikes in 30 min of recording. Duplicate spike trains were identified by temporal cross-correlation and removed.

## Visual display

Visual stimulation was performed with the optically reduced image from a computer display. In 14 experiments a gamma-corrected OLED microdisplay (eMagin, Bellvue, WA) refreshing at 60.35 Hz was used. In one experiment a gamma correct CRT display (Sony Trinitron) refreshing at 120 Hz was used. The image was focused through the retina onto the photoreceptor outer segments. The emission spectrum of each display primary was measured with a spectroradiometer (PR-701, PhotoResearch, Chatsworth, CA) after passing through the optical elements between the display and the retina. The mean photoisomerization rates for the L, M and S cones were estimated by computing the inner product of the power scaled emission spectra per unit area with the spectral sensitivity of each opsin, and multiplying by the effective collecting area of primate cones (0.37 $\mu m^2$) (*Baylor et al., 1987*). The power of each display primary was measured at the preparation with a calibrated photodiode (UDT Instruments, San Diego, CA). At the mean background illumination level of the OLED display, the photoisomerization rates for the rods, L, M, and S cones were approximately 29,120, 9440, 9270, and 2320 photoisomerizations per receptor per second (4670, 2200, 2200, 900 for the CRT display). These estimates were not corrected for the angle of illumination and pigment self screening in the cone outer segments, because the precise angle of illumination and the amount of bleached pigment were unknown.

## Receptive field measurements

The spatial, temporal and chromatic response properties of recorded RGCs were characterized using a movie stimulus composed of an array of square pixels whose values were updated randomly and independently on each frame (see *Chichilnisky, 2001*). For initial characterization with large pixels, the intensity of each display primary at each pixel location was independently chosen from a binary distribution, yielding a stimulus with chromatic variation. For high resolution characterization of individual cones, the intensities of the display primaries were modulated in unison in most experiments (13 of 15). This yielded a black-white binary intensity modulation with higher variance and thus greater modulation of RGC responses. In the other two experiments, the display primaries were modulated independently of one another. In both conditions, the contrast of each primary (difference between the maximum and minimum intensities divided by the sum) was 96%. For low spatial resolution receptive field maps used for classification (not shown), the pixels were 25.5 or 34 μm on a side, the stimulus refresh rate was 20, 30, or 60 Hz, and recording duration was 30 min. For high resolution maps (*Figures 1, 3, 4*), the pixels were 3.4 μm on a side, the stimulus refresh rate was 10, 12, or 15 Hz, and recording duration was 30–60 min (depending on the experiment). High resolution spatial receptive fields shown in *Figures 1, 3, 4* were calculated by collapsing the spike-triggered average over time. For this purpose, the time course was calculated from the average of a subset of stimulus pixels whose absolute peak intensity exceeded four robust standard deviations of all pixel intensities.

## Identification of OFF midget RGCs

RGC classification was performed as described previously (*Field et al., 2007*; *Petrusca et al., 2007*; *Field et al., 2010*). OFF midget RGCs were readily identified as those cells with the smallest receptive fields of the appropriate sign, and relatively sustained temporal response properties. The identification of distinct cell types was confirmed by the mosaic organization of receptive fields within each type, and the identities of the different types were consistent with anatomical density measurements (*Chichilnisky and Kalmar, 2002*).

## Cone finding and preprocessing

Cone finding was performed using a Bayesian approach described previously (*Field et al., 2010*). Briefly, the linear spatial receptive field of each cone was modeled as a gaussian, and the linear spatial receptive field of each RGC was modeled as a sum of cones. For each retina, a mosaic of cones was greedily chosen to maximize the likelihood of spiking responses across all reliably identified cells (including multiple types). After identifying cone locations, the high-resolution noise stimulus was integrated against the Gaussian profile of each cone, to obtain a set of cone input signals for each RGC. These signals were then convolved in time with the temporal component of the spike-triggered average, effectively aligning the stimulus with the response in time. All modeling subsequently

focused only on the spatial component of the response. Spikes were binned by the timing of stimulus frames, which across experiments was typically 12 Hz (83 ms bins). This spatial and temporal preprocessing yielded, for each RGC, a $C \times T$ stimulus matrix $X$, where $C$ is the number of cones providing input to that RGC, and $T$ is the number of stimulus frames in the experiment, and a $T \times 1$ vector $y$ of spike counts. These preprocessing steps assume a linear model of the ganglion cell response, but linear analysis is adequate for the purpose of cone localization, because it only requires that each cone provides non-zero input to at least one ganglion cell, which will be true for most monotonic non-linearities; the same procedure was also anatomically validated in previous work (*Field et al., 2010*).

## Model formulation

The subunit model describes the transformation of cone signals to the spiking response of a RGC using two stages of computation. Formally, the complete model is a linear-nonlinear-linear-nonlinear-poisson cascade, generalizing and extending both LNP models (*Simoncelli et al., 2004*; *Pillow et al., 2008*), models with 'input nonlinearities' (*Ahrens et al., 2008*; *McFarland et al., 2013*), and previous subunit models (*Rust et al., 2005*; *Vintch et al., 2012*; *Lochmann et al., 2013*). The first stage linearly combines cone inputs into 'subunits' followed by a instantaneous subunit nonlinearity. Let $U$ be the $S \times C$ weight matrix connecting $C$ cones to $S$ subunits. $U$ is partitioned as $I \odot A$, where $I$ is a binary indicator matrix, $A$ is a continuous-valued weight matrix, and $\odot$ is the pointwise product. $I$ specifies which cones provide inputs to which subunits, and $A$ captures the weight on each cone within a subunit. The response of subunit $s$ at time $t$ is

$$y_{st} = f\left( \sum_{c=1}^{C} u_{sc} x_{ct} \right),$$

where $f(\bullet)$ is the subunit nonlinearity (see below for parameterization), and $x_{ct}$ and $u_{sc}$ are elements of the matrices $X$ and $U$. The second stage of the model linearly combines subunit responses followed by an instantaneous output nonlinearity. For any given RGC, let $w$ be the $1 \times S$ vector encoding the weights from subunits to that RGC. The response (mean firing rate) of the RGC at time $t$ is

$$z_t = g\left( \sum_{s=1}^{S} w_s y_{st} \right),$$

where $g(\bullet)$ is the output nonlinearity and $w_s$ is an element of the vector $w$. The nonlinearities $f$ and $g$ were parameterized as piece-wise polynomials (specifically, cubic splines) with eight node points. Our optimization procedure sought parameters $I$, $A$, $w$, $f$, $g$ to maximize the log likelihood of the measured spike counts $r_t$, assuming that the number of spikes in at each time $t$ was given by a Poisson distribution with rate parameter $z_t$:

$$\log p(r_t | z_t(x_{ct}; I, A, w, f, g)) = \sum_t r_t \log z_t(x_{ct}; I, A, w, f, g) - \sum_t z_t(x_{ct}; I, A, w, f, g)$$

## Parameters and constraints

The dimensionality of the parameter space is substantial, and the objective function is non-convex and likely to have many local minima. To encourage reliable estimation, while remaining sufficiently flexible to capture a variety of nonlinear RGC properties, we introduced several constraints: (1) Nonlinearity smoothness was enforced through a set of linear constraints on the spline parameters that guarantees first and second-order derivatives are defined at the nodes (implying that the output of a subunit is smoothly related to its input with no discontinuities); (2) The binary assignment matrix $I$ was assumed to be orthogonal (i.e., the subunits were non-overlapping): each cone provided input to exactly one subunit, and the number of subunits was thus constrained to lie between one (equivalent to an LN model) and the number of cones (which we refer to as the single-cone subunit model). (3) The weights from cones to subunits were assumed to be positive, and the weights within each subunit were assumed to sum to 1. No constraints were imposed on the sign or magnitude of weights in the second stage.

## Model estimation

Even with constraints, the two-stage structure of the model, and the combination of binary assignments and continuous-valued weights (and nonlinearities) makes optimization difficult. We decomposed the problem into a greedy search over the assignments, and for each assignment performed an optimization over the continuous-valued parameters.

The optimization over the continuous-valued parameters—the cone weights ($A$), the subunit weights ($w$), and the nonlinearities ($f$, $g$)—was performed using an alternating sequence of simpler gradient descent problems (known as 'coordinate decent'). First, $g$ was initialized as $\log(1 + \exp(\bullet))$, $f$ was initialized as negative half-wave rectification, $w$ and $A$ were initialized to contain values of 1. These parameters were then repeatedly optimized, sequentially, in the following order: $w$, $A$, $f$. The nonlinearity $g$ was only refit once at the end. In practice, we found that repeating the full optimization after estimating $g$ did not substantially change the recovered parameters. Most of these subproblems correspond closely to estimation problems that have been characterized in previous literature on fitting neural encoding models (*Chichilnisky, 2001*; *Paninski, 2004*; *Ahrens et al., 2008*). For example, if the weights ($A$) and nonlinearities ($f$, $g$) are held fixed, estimating the subunit weights ($w$) is identical to fitting a single-stage LNP model, which can be done with gradient descent as long as $f$ is convex and log-concave (*Paninski, 2004*). In practice, each of the subproblems on their own, as well as the combined procedure, converged to unique solutions from random initial starting points.

The best-fitting assignment matrix $I$ was found through a greedy procedure. $I$ was initialized with the identity matrix (the 'single cone' subunit model), all remaining model parameters were estimated (as described above) and the maximized likelihood recorded. For every possible merger of two subunits into a larger subunit, all the remaining model parameters were re-estimated (as described above), and the maximized likelihood recorded. If any of the potential merged pairs yielded an improvement in likelihood over that of the unmerged model, those two subunits were merged. This process was repeated, at each step considering a merger of all possible pairs of subunits, until none of the pairings offered an improvement in likelihood. All code used for model fitting is available at https://github.com/freeman-lab/subunits and an interactive web visualization is available at https://github.com/freeman-lab/subunits-web.

## Validation of model fitting procedure on simulated data

Simulations were performed to confirm that the estimation procedure reliably recovered model parameters. Spiking responses were generated using a known set of parameters, with values chosen as typical for those recovered from RGCs, and simulation duration was matched approximately to that of typical experiments (as well as smaller and larger values). The model was fit to these simulated responses. The recovered parameters were found to be nearly identical to the parameters used for simulation (*Figure 3—figure supplement 1*).

## Alternative models

The accuracy of the subunit model was compared to a simpler LN model, containing only weights on each cone and an output nonlinearity. Formally, the subunit model reduces to the LN model if $I$ and $A$ are identity matrices and $f$ is linear, leaving the parameters $w$ and $g$. (An alternative parameterization of LN behavior would be to make $I$ a vector of ones, i.e. put all cones in a single subunit). In some cases the subunit model was also compared to a subunit model that included a subunit nonlinearity ($f$), but with an identity matrix for $I$ and $A$; that is, the single cone subunit model. In all cases, parameters were fit using similar likelihood maximization methods to those used to fit the full subunit model.

## Assessing model accuracy

For every RGC, the model was fit to 80% of the data, and the accuracy of the model was evaluated on the remaining 20%. To account for possible nonstationarities in responses, the subsets of data used for fitting were drawn randomly from periods of time distributed throughout the duration of the experiment. On the evaluation data, model performance was evaluated as

$$R^2 = 1 - \frac{\sum_t (z_t - r_t)^2}{\sum_t (r_t - \overline{r_t})^2}$$

Although the model was fit by maximizing log likelihood, we used the more intuitive metric of $R^2$ to report performance. Comparisons between models were similar when using log likelihood. For repeated white noise and grating data (*Figures 5, 6*), $R^2$ was adjusted to account for the reliability of the responses (see below). For each retina, the improvement of one model over another (e.g., subunit over LN) was estimated as the slope of the best-fitting regression line (through the origin) relating the $R^2$ of the two models across RGCs (i.e., the best-fitting line to the scatter plots in *Figure 3C,D*). In a small number of cases, when analyzing the maximum differentiated stimuli (see below), $R^2$ for the LN model was negative, due to particularly poor predictions on these stimuli. In all such cases, the performance of the subunit model was positive. To avoid the influence of these outliers on average performance improvement, they were excluded from the calculation.

## Maximally differentiated stimuli

To better differentiate the models, stimuli were selected on the basis of the model predictions. Specifically, the models were fit to training data, and then used to generate predicted responses for stimuli from the remaining data. From amongst these held out stimuli, we selected the 20% individual stimulus frames for which the squared difference in the prediction of the two models was largest. Selecting stimuli by individual frames was appropriate because stimuli were preprocessed to have a one-to-one correspondence with the response at each frame. Performance of both models was then compared to actual neural response on these stimuli. This selection does not favor either model a priori, because it is not based on their performance in explaining the data, but only on how much their predictions differ.

## White noise repeats

For the results of *Figure 4*, the model was fit to a single run of independent white noise, and then tested on a 10 s sequence of white noise that was repeated 100 times. The repeated presentations allowed prediction accuracy to be computed relative to the inherent reliability of the responses. We compared model predictions of the spike count on individual trials of each 10 s stimulus sequence to the actual responses averaged over all remaining runs of the sequence. In each case, the performance was quantified as the square of the correlation coefficient. This is similar to $R^2$ as defined above, but allows for an arbitrary offset and gain, which was important for some RGCs due to changes in overall firing rate between the two experimental preparations. These two $R^2$ values were averaged across all trials, and their ratio yielded an adjusted $R^2$, representing the fraction of explainable variance accounted for by the model.

## Gratings

Contrast reversing gratings modulating at 2 Hz were presented at eight spatial phases for each of 10 spatial periods. Each presentation of a grating lasted 8 s followed by 2 s of a uniform screen that had an intensity equal to the mean grating intensity over time. Each grating of a particular spatial period and phase was presented three times. For a RGC, the mean spike rate over one contrast reversal was estimated by averaging responses over the 16 contrast reversals (8 s × 2 Hz) in each of these three grating presentations (*Figure 6A*). The grating spatial periods ranged from 10 to 1305 microns on the retina. Data reported here (*Figure 5*) only for the five spatial periods matched to (or smaller than) a typical OFF midget RGC receptive field. As for white noise, model accuracy was evaluated as adjusted $R^2$. For each spatial period and phase, the response to each of the 36 repeated presentations of the grating (16 contrast reversals × 3 repeats) was predicted by (1) the model and (2) the average across other presentations, and their ratio (averaged across trials) yielded an adjusted $R^2$.

## Single and paired cone stimulation

Cone locations and subunit model fits were obtained online during experiments, computed in parallel across roughly 10 multi-core Linux workstations. With parallelization, hundreds of RGCs from an entire retina could be fit in under 10 min. Once model fits were obtained for every RGC, several RGCs (5–10) with high SNR and a variety of subunit configurations were chosen for targeted single cone and paired cone stimulation. Chosen RGCs were at least two cell spacings away from each other, to minimize the impact of stimulating cones lying in the receptive field surround of any of the chosen RGCs. Typically four cones were chosen per RGC. For simplicity, *Figure 7* only reports data for the three cones per RGC showing the strongest response to single-cone stimulation, but the quantitative analysis (see below) included all cones tested. Cone location analyses generated stimulus screen coordinates for the center of every cone found in the mosaic. Coordinates of cones within the receptive field of the chosen RGCs were then used to specify regions of the screen selected to stimulate single cones, as follows. Depending on cone spacing in each preparation, spots of radius 7.65–9.35 µm were generated around each cone center location. The full native screen resolution was used (pixels 1.7 µm on a side). For cases in which cone spacing was close enough that the resulting regions would overlap, pixels were assigned to whichever competing cone had the nearest center coordinates. Single cones were then stimulated with uniform contrast steps over the defined regions. Recording trials were 750 ms long with the stimulus presentation occurring in the first 250 ms. Stimulation region (i.e., choice of cone) and contrast were randomized across trials. Control experiments in which light stimuli were presented within the spaces in between cones demonstrated the specificity of this stimulation (*Li et al., 2014*).

## Accuracy of closed-loop increment-decrement predictions

For each RGC, each selected cone was presented with an increment or decrement of light alone, and then separately paired with an increment of light to three other cones. For each such selected cone, the firing rate response to single or paired stimulations were averaged across repeated trials (typically 20–40) and within a 300 ms time window following stimulus presentation. These four firing rates were then divided by their maximum. This normalization served to emphasize the relative effect of pairing increments with a decrement of light, as well as correct for differences in effective cone strength between measurement and model due to, for example, small movement of the retina in between stimulus presentations. For each RGC, model predictions were generated by providing negative or positive inputs to the same combination of cones tested in the experiment. The predictions were then similarly divided by the maximum predicted firing rate for each cone. Finally, the predictive accuracy was computed as the square of the correlation coefficient between predicted and measured response across all cone pairs. Given the non-normal estimator of interest (a difference of correlation coefficients), a bootstrap test was used to compare predictions of the different models—full subunit, single-cone subunit, and LN. On each iteration of a bootstrap, cone pairs were randomly sampled with replacement across RGCs, accuracies for all models were reestimated (as above), and the differences in accuracy between models were computed. The accuracies were considered significantly different if the fifth percentile of the resulting distribution of differences exceeded 0.

# Additional information

### Funding

| Funder | Grant reference | Author |
| --- | --- | --- |
| Howard Hughes Medical Institute (HHMI) | | Eero P Simoncelli, Jeremy Freeman |
| National Eye Institute (NEI) | EY018003 | Liam Paninski, Eero P Simoncelli, EJ Chichilnisky |
| National Eye Institute (NEI) | EY017992 | EJ Chichilnisky |
| Whitehall Foundation | | Greg D Field |

The funders had no role in study design, data collection and interpretation, or the decision to submit the work for publication.

## Author contributions

JF, EPS, Conception and design, Analysis and interpretation of data, Drafting or revising the article; GDF, EJC, Conception and design, Acquisition of data, Analysis and interpretation of data, Drafting or revising the article; PHL, Conception and design, Acquisition of data, Analysis and interpretation of data; MG, DEG, KM, AS, AML, Acquisition of data; LP, Analysis and interpretation of data

## Author ORCIDs

Greg D Field, http://orcid.org/0000-0001-5942-2679
Peter H Li, http://orcid.org/0000-0001-6193-4454
EJ Chichilnisky, http://orcid.org/0000-0002-5613-0248

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
