## [Decision Letter]

Thank you for sending your work entitled “Mapping nonlinear receptive field structure in primate retina at single cone resolution” for consideration at *eLife*. Your article has been favorably evaluated by Eve Marder (Senior Editor) and three reviewers, one of whom, Matteo Carandini, is a member of our Board of Reviewing Editors.

The Reviewing editor and the other reviewers discussed their comments before we reached this decision, and the Reviewing editor has assembled the following comments to help you prepare a revised submission.

This useful and interesting paper presents new modeling methods for characterizing retinal ganglion cell receptive fields, and applies it to explain the nonlinear spatial summation in primate OFF midget ganglion cells. The results are surprising, since midget cells (which constitute the vast majority of retinal ganglion cells in primates) were hitherto thought to operate mostly linearly, whereas here they are described as being fundamentally nonlinear. The data are of high quality, the modeling thoughtful and innovative, and the paper clear and well-presented. Some questions, however, need to be addressed.

Description of the model: Even though the paper is written for a general audience, the modeling should be explained more extensively. Currently, one has trouble understanding the cone finding and preprocessing section (Methods). The paper suggests that the cone positions and weights were fit using previously described methods. Then it states that the signals were convolved with the ganglion cell's spike-triggered average (STA). But the STA is derived from a linear model of the response. Why is this appropriate for a step in building the subunit model? A similar concern: the initial maps of the cones are derived from a linear analysis (STA), and so wouldn't these maps potentially miss nonlinear sources of the response?

Relationship to known nonlinearities in midget cells: It is interesting to see such pronounced nonlinearities (subunits) in the receptive fields of a class of retinal ganglion cells (RGCs), the midget cells, which are commonly described as linear (X-like). The paper should do a better job at casting these nonlinearities in the context of the existing knowledge, to point out whether they are consistent with previous findings, or whether they are a surprise. For instance, the paper could describe more extensively the prior literature on linearity/nonlinearity of P cells in the LGN, including several papers by Benardete and Kaplan, by Derrington and Lennie, and at least one by the Movshon laboratory ([43], which is cited in the paper). To what extent are those findings consistent with the findings here? (This applies also to differences and similarities between ON and OFF P cells, see next point).

Focus on OFF cells: The paper focuses on OFF midget cells, which apparently had the highest signal-to-noise ratio and stability. But what about ON cells? Are they ignored because they are actually linear? In that case, it would help to start the paper with those, and demonstrate linearity. Then go on to the OFF ones and show the nonlinearity. If instead they have more complex nonlinearities, it would help to at least describe them, ideally with a figure.

Nature of the large subunits: The reported presence of such large photoreceptor subunits (in contrast to preexisting literature) deserves to be given more prominence, to be discussed more extensively, and ideally to be investigated experimentally. A possible explanation for such large subunits could lie in gap-junction among photoreceptors. Ideally one would test this possibility experimentally, e.g. by adding dopamine D1 agonist or antagonist to modulate the gap-junctional link among photoreceptors (or at least test with the model if cones that are associated in one subunit for an OFF RGC tend to be associated as one subunit for an overlapping ON RGC?). Another possible explanation lies in amacrine cells feeding signals from surrounding bipolar cells. These cells are currently not considered in the model, and might result in single subunits pooling from very large photoreceptors sets in the model. Another possibility, which is not possible to rule out given the way the model is explained, is that some sort of constraint (or minimization) is posed on the number subunits (bipolar cells), leading to artifactually large subunits. Finally, if data are available from a region closer to the fovea, it would be very interesting to see whether the model is able to capture the relationship between 1 photoreceptor – 1 bipolar cell – 1 retinal ganglion cell.

Number of subunits: Another result of the model that must be compared with morphological literature (either [79] or [41]) is the number of subunits (bipolar cells). This subject is pivotal to understand how the results resemble primates' anatomy. This issue deserves a Figure similar to Figure 6 and an articulated discussion.

Modeling of nonlinearities: The subunit nonlinearity seems to be modeled as being static, and being the same for all subunits. Any evidence that this is a good assumption? Moreover, can the fits be further improved by imposing a temporal nonlinearity (not just a static one)? For example, direct measurements of glutamate release (imaging of iGluSnFR) in ON layers, revealed that rectification is mild, and that a temporal nonlinearity (transient increases in release) explained nonlinear behavior in postsynaptic ganglion cells (8). The same might be true in primate midget cells.

In the second paragraph of the subsection “Nonlinear spatial summation in OFF midget RGCs”, the authors should demonstrate the phase dependence of the OFF midget cell responses. This indeed does not follow the rule for Y-like cells and may give insights into the underlying circuitry for generating the nonlinearity. Does the subunit model show similar behavior as the recorded cell? Later in the paragraph, the authors raise the possibility that crossover inhibition may generate differences between midget and parasol cells. This suggestion needs more context/explanation.

Single-neuron approach: The way the data are presented, these RGCs could have been measured one at a time. And instead, they were measured all together. This is a missed opportunity. For instance, if some cones are arranged in a subunit for one RGC, does that increase their chance of being arranged as a subunit in another RGC? Along the same lines, does the model help in predicting correlations between different RGCs?

Relationship with measurements in vivo: The authors take the interesting viewpoint that their measurements in vitro are actually closer to being natural than measurements under anesthesia, which may affect the basal release rate and therefore the strength of nonlinearity at the set point. This is a key issue, and if one wanted to prove it one could perhaps add the same anesthetics to the in vitro retina as used during in vivo recordings. The alternative explanation is that it is the in vitro condition that somehow distorts the nonlinearity. The washout of neuromodulators could be a source of variability in ex vivo preps, too. In the absence of experimental data, this issue needs to be treated in a more balanced way, and both possibilities kept open.

Room for improvement: It would be useful to better describe room for improvement in the model, if possible. For example, the data in the last figure apparently were difficult to capture with the subunit model (low R2 values), although the model fits in this case were not shown. It would be useful to show both subunit and linear model fits in this figure. Also, a surprising omission from the model is inhibitory cells (e.g., amacrine cells) pooling over larger areas. Is there any evidence that data would be better fit by including them in a model?

Minor comments:

The paper states repeatedly that the improvement in predictions made by the model over the LN model is minor. Yet, in responses to gratings it seems major (Figure 5). This might be worth emphasizing.

For instance, when predicting the responses to gratings (Figure 5), it would help to see a plot with spatial frequency in the abscissa, and harmonic responses – both first and second – in the ordinate. And curves predicted by the two models.

---

## [Author Response]

This useful and interesting paper presents new modeling methods for characterizing retinal ganglion cell receptive fields, and applies it to explain the nonlinear spatial summation in primate OFF midget ganglion cells. The results are surprising, since midget cells (which constitute the vast majority of retinal ganglion cells in primates) were hitherto thought to operate mostly linearly, whereas here they are described as being fundamentally nonlinear. The data are of high quality, the modeling thoughtful and innovative, and the paper clear and well-presented. Some questions, however, need to be addressed.

Description of the model: Even though the paper is written for a general audience, the modeling should be explained more extensively. Currently, one has trouble understanding the cone finding and preprocessing section (Methods). The paper suggests that the cone positions and weights were fit using previously described methods. Then it states that the signals were convolved with the ganglion cell's spike-triggered average (STA). But the STA is derived from a linear model of the response. Why is this appropriate for a step in building the subunit model? A similar concern: the initial maps of the cones are derived from a linear analysis (STA), and so wouldn't these maps potentially miss nonlinear sources of the response?

We agree that the description of the modeling was a bit unclear on these points. The simple answer is that the STA procedure is used *only* as a preprocessing step to estimate the location and size of the cones. Once this is completed, the stimuli are linearly transformed from pixel arrays to the lower-dimensional space of cone excitation values, and these are used for the model-fitting (which determines the detailed nonlinear contribution of the cones to the ganglion cell response). A simple linear STA analysis is adequate for the purpose of cone localization because it relies only on each cone providing, on average, a non-zero input to the response of at least one ganglion cell. While there are some classes of subunit nonlinearities that could in principle cause cones to be missed in the STA calculation (e.g. full-wave rectification), in practice most monotonic nonlinearities, including half rectification, which appears to dominate in midget cell responses, will not do so. Note that the [58] paper developed this method, and showed that the estimated locations were in close correspondence with anatomical measurements. The use of the STA to estimate cone size/extent is justified because each cone combines the pixel intensities within its aperture approximately linearly ([62], Visual Transduction of Cones in the Monkey). Finally, convolving with the temporal component of the STA primarily served only to align stimulus and response times, and all subsequent aspects of the analysis focused entirely on the spatial component of the response.

We have clarified and expanded these points in the Methods.

*Relationship to known nonlinearities in midget cells: It is interesting to see such pronounced nonlinearities (subunits) in the receptive fields of a class of retinal ganglion cells (RGCs), the midget cells, which are commonly described as linear (X-like). The paper should do a better job at casting these nonlinearities in the context of the existing knowledge, to point out whether they are consistent with previous findings, or whether they are a surprise. For instance, the paper could describe more extensively the prior literature on linearity/nonlinearity of P cells in the LGN, including several papers by Benardete and Kaplan, by Derrington and Lennie, and at least one by the Movshon laboratory (*[43]*, which is cited in the paper). To what extent are those findings consistent with the findings here? (This applies also to differences and similarities between ON and OFF P cells, see next point).*

We agree this is a very important issue. Although the reviewers are presumably advocating for treating the issue in Results, we chose to handle it in the Discussion and continue to favor doing so. There is a developed literature on midget/P-cells and their similarity to cat X-cells. We have reviewed this literature and find that most previous in vivo studies measured frequency doubling in P-cells at eccentricities <10 deg (e.g. [39] and [43]). At these eccentricities, the vast majority of macaque-midget RGCs receive input from just one cone (41). With a single cone in the receptive field, there is no possibility of a spatial (Y-like) non-linearity in the receptive field center, because the cone integrates light linearly over its collecting aperture ([62], Visual Transduction of Cones in the Monkey). One previous in vivo study in which P-cells were studied over a larger range of eccentricities (0-30 deg) did find midget cells with a substantial second harmonic response to contrast reversing gratings (23). Although this point was not strongly emphasized in the Abstract, it is clear in the data (Figures 2 and 6). The authors did not analyze the magnitude of the second harmonic as a function of eccentricity, but their results appear consistent with those presented here as well as other previous ex vivo studies from the peripheral primate retina (Petrusca et al. 2006, [12]). Our interpretation is that the apparent discrepancy probably results from the fact that midget cells appear X-like when they sample from only one cone, while spatial non-linearies are revealed at eccentricities where they sample from multiple cones. In other words, there is no factual disagreement with the literature, and our results point to a possibly nearly ubiquitous mechanism in RGCs. This point is discussed in detail in the second paragraph of the Discussion.

Focus on OFF cells: The paper focuses on OFF midget cells, which apparently had the highest signal-to-noise ratio and stability. But what about ON cells? Are they ignored because they are actually linear? In that case, it would help to start the paper with those, and demonstrate linearity. Then go on to the OFF ones and show the nonlinearity. If instead they have more complex nonlinearities, it would help to at least describe them, ideally with a figure.

The ON-midget cells do not behave linearly. We have found that they also exhibit nonlinear spatial integration, and that the model presented here provided a more accurate prediction of their response to white noise and gratings than an LN model. However, we also find that they exhibit additional complexities within their receptive field that were not observed in OFF-midget cells. Namely, a decrement of light presented to one cone in the receptive field center enhances the response to a simultaneously presented increment presented to another cone (this was not found in any OFF midget cells). We are currently investigating the possibility that this is caused by a RF surround within the subunit, facilitating the response of the RF center of the subunit. To explore this issue fully will require a more sophisticated model, novel fitting approaches, and a collection of additional experiments. Thus we view it as beyond the scope of this manuscript. We feel it would be premature to report the findings on ON-midget cells without a fuller exploration, so we would prefer to leave the above comments out of the manuscript.

Nature of the large subunits: The reported presence of such large photoreceptor subunits (in contrast to preexisting literature) deserves to be given more prominence, to be discussed more extensively, and ideally to be investigated experimentally. A possible explanation for such large subunits could lie in gap-junction among photoreceptors. Ideally one would test this possibility experimentally, e.g. by adding dopamine D1 agonist or antagonist to modulate the gap-junctional link among photoreceptors (or at least test with the model if cones that are associated in one subunit for an OFF RGC tend to be associated as one subunit for an overlapping ON RGC?). Another possible explanation lies in amacrine cells feeding signals from surrounding bipolar cells. These cells are currently not considered in the model, and might result in single subunits pooling from very large photoreceptors sets in the model. Another possibility, which is not possible to rule out given the way the model is explained, is that some sort of constraint (or minimization) is posed on the number subunits (bipolar cells), leading to artifactually large subunits. Finally, if data are available from a region closer to the fovea, it would be very interesting to see whether the model is able to capture the relationship between 1 photoreceptor – 1 bipolar cell – 1 retinal ganglion cell.

We thank the reviewers for this comment. There are several possible mechanistic underpinnings of the large receptive field subunits observed for some cells (we assume that the reviewers are referring to Figure 6). The reviewers suggest testing the role of gap junctions by modulating cone-cone gap junctions with D1 agonist or antagonists. This is an interesting idea and one that is worth pursuing, but it requires an extensive repeat of all the experiments shown here, with the additional complexity of pharmacology and controls, and we view it is beyond the scope of this study. First, it is not clear that dopamine modulates cone-cone gap junctions in the primate retina. Dopamine does not appear to modulate rod-cone gap junctions in the primate retina (Schneeweis and Schnapf 1999) as has been found in lower vertebrates. Second, both D1-like and D2-like receptors have been localized to primate photoreceptors (Zarbin 1986; Dearry 1991), so it is unclear which should be targeted. Finally, dopamine has effects at many locations in the retina (Witkovsky 2004). Knowing how to conclusively interpret the results of manipulating dopamine in the primate retina would require extensive intracellular recordings from photoreceptors, bipolar cells and ganglion cells. In short, understanding the role of dopamine receptor signaling among photoreceptor coupling in the primate retina is a major endeavor.

The reviewers also suggest analyzing ON and OFF midget cells with overlapping receptive fields; subunit structure that is caused by cone-cone gap junctions would be expected to be common to these two pathways. This is a good suggestion, but (1) as explained above, we’ve elected to leave ON cells out of the paper, and (2) obtaining a data set with relatively complete mosaics of ON and OFF cells is quite difficult (some of the difficulty is due to spike sorting issues). We’d prefer not to delay publication of the current results, which are already well developed and warrant extensive documentation.

The reviewers also suggest the possibility of amacrine cells. In general, the ∼40 types of amacrine cells in the mammalian retina are considered to be inhibitory in their interaction with bipolar and ganglion cells. It is not clear how they would generate the excitatory subunit structure observed here. However we have added a sentence to the Discussion to acknowledge this possibility.

Finally, the reviewers also ask whether a constraint on the minimum number of subunits might lead to an overestimation of subunit size. In our model fitting, there is no constraint on subunit structure other than disallowing subunit overlap (each cone belongs to one subunit). In fact, in many retinas and RGCs, there is a one-to-one correspondence between cones and subunits, which means the model finds as many subunits as justified through an increase in explanatory power. Unfortunately, we do not have data from sufficiently close to the fovea to capture the 1 cone – 1 subunit – 1 RGC phenomenon suggested by the reviewers, because the size and density of cells and spikes near the fovea makes spike sorting very difficult.

*Number of subunits: Another result of the model that must be compared with morphological literature (either*
[79]
*or*
[41]*) is the number of subunits (bipolar cells). This subject is pivotal to understand how the results resemble primates' anatomy. This issue deserves a Figure similar to*
Figure 6
*and an articulated discussion.*

We thank the reviewers for this comment. Midget bipolar cells dendrites are believed to tile the retina with a coverage factor of 1, and their dendrites do not overlap ([79], see also Wassle 2009). Thus the number of subunits is determined by the convergence of cones to bipolar cells (the statistic that we report in Figure 6), and reporting the number of subunits would be redundant with reporting the cone convergence to each subunit. We have added a sentence to the Results to clarify this important point.

*Modeling of nonlinearities: The subunit nonlinearity seems to be modeled as being static, and being the same for all subunits. Any evidence that this is a good assumption? Moreover, can the fits be further improved by imposing a temporal nonlinearity (not just a static one)? For example, direct measurements of glutamate release (imaging of iGluSnFR) in ON layers, revealed that rectification is mild, and that a temporal nonlinearity (transient increases in release) explained nonlinear behavior in postsynaptic ganglion cells (*[8]*). The same might be true in primate midget cells.*

This is an interesting suggestion and worthy of further investigation. It is likely that including temporal nonlinearities would further improve model performance, at the expense of a significant increase in free parameters and fitting complexity. Unfortunately, this would require collecting substantially longer data sets than the ones obtained in this paper. Our goal is to provide an extension of the classical “LNP” model that can be fit to responses to white noise stimuli and that phenomenologically accounts for nonlinear spatial summation. However, this point is important, so we have added a sentence to the Discussion to point out that dynamic nonlinearities likely also play a role in shaping RGC responses.

In the second paragraph of the subsection “Nonlinear spatial summation in OFF midget RGCs”, the authors should demonstrate the phase dependence of the OFF midget cell responses. This indeed does not follow the rule for Y-like cells and may give insights into the underlying circuitry for generating the nonlinearity. Does the subunit model show similar behavior as the recorded cell? Later in the paragraph, the authors raise the possibility that crossover inhibition may generate differences between midget and parasol cells. This suggestion needs more context/explanation.

Thank you for this suggestion. The example plots in Figure 5 suggest some degree of phase dependence in the response. We tried to further quantify this by computing F1 and F2 amplitude for the four presented phases. Although measures of F1 and F2 for both models and data were interpretable (and are reproduced in Figure 8, in reply to another comment), responses after splitting up by phase were noisy and difficult to interpret (likely due to the small number of repeats per phase), so we have opted not to include these analyses.

Single-neuron approach: The way the data are presented, these RGCs could have been measured one at a time. And instead, they were measured all together. This is a missed opportunity. For instance, if some cones are arranged in a subunit for one RGC, does that increase their chance of being arranged as a subunit in another RGC? Along the same lines, does the model help in predicting correlations between different RGCs?

This is a really interesting point, but may be more applicable to other RGC types. Within each midget cell type (ON and OFF), the bipolar axon terminals form a mosaic, and the receptive fields of neighboring midget RGCs show very little overlap. So there is very little opportunity for shared subunits and indeed we have not seen any to date. Between ON and OFF midget cells, we wouldn’t expect to find shared subunits because they receive input from different bipolar types (ON and OFF midget bipolars, respectively).

Relationship with measurements in vivo: The authors take the interesting viewpoint that their measurements in vitro are actually closer to being natural than measurements under anesthesia, which may affect the basal release rate and therefore the strength of nonlinearity at the set point. This is a key issue, and if one wanted to prove it one could perhaps add the same anesthetics to the in vitro retina as used during in vivo recordings. The alternative explanation is that it is the in vitro condition that somehow distorts the nonlinearity. The washout of neuromodulators could be a source of variability in ex vivo preps, too. In the absence of experimental data, this issue needs to be treated in a more balanced way, and both possibilities kept open.

Our intention was not to take a stand that either ex vivo or anesthetized in vivo measurements are closer to being natural, but only to point out that it is difficult to know which is closer to natural, and that more work will be required to determine this (as the reviewers state). We have clarified our language on this issue with new text.

Room for improvement: It would be useful to better describe room for improvement in the model, if possible. For example, the data in the last figure apparently were difficult to capture with the subunit model (low R2 values), although the model fits in this case were not shown. It would be useful to show both subunit and linear model fits in this figure. Also, a surprising omission from the model is inhibitory cells (e.g., amacrine cells) pooling over larger areas. Is there any evidence that data would be better fit by including them in a model.

Inhibitory inputs and RF surrounds were not included in the modeling work here. As the reviewers point out, our goal was to develop a model and fitting procedure that capture excitatory spatial subunit structure within the receptive field center. The surround, and particularly the diverse array of amacrine cell types, are of great interest to a deeper understanding retinal processing. But, given their diversity and largely unknown properties, to include their potential effects would require a substantial new set of data, analysis, and modeling. We have added text to the Discussion to make this point and to present clearly all the directions in which the modeling could be extended: neighbor-neighbor interactions, temporal nonlinearities, post-spike filter, RF surround, and inhibitory inputs. The predictive accuracies the reviewers mention (from Figure 7) are only moderately lower than that obtained for typical midget cells (e.g. Figure 3); that difference may be due to the relatively stronger increment-decrement used for evaluating the model, compared to the white noise used to train it, as is now suggested in the text.

Minor comments:

*The paper states repeatedly that the improvement in predictions made by the model over the LN model is minor. Yet, in responses to gratings it seems major (*Figure 5*). This might be worth emphasizing.*

We didn’t want to overstate the results, but we agree that we could probably be more clear on this point. We have tried to balance the description of the result with new text (subsection “Subunit model more accurately predicts responses to grating stimuli”).

*For instance, when predicting the responses to gratings (*Figure 5*), it would help to see a plot with spatial frequency in the abscissa, and harmonic responses – both first and second – in the ordinate. And curves predicted by the two models.*

We generated the figure suggested by the reviewers (please see Figure 8; for three example cells, dashed lines are F2, solid lines are F1, purple is subunit model prediction, green is LN model prediction). These representations of the data confirm that the F1 drops off with spatial frequency whereas the F2 does not, and the subunit model is broadly consistent with this behavior whereas the LN model is not. Although informative, this representation does not add fundamentally to our conclusions, so we would prefer not to add it to the manuscript.

Author response image 1.**DOI:**
http://dx.doi.org/10.7554/eLife.05241.013